# Enhancing Solutions for Complex PDEs: Introducing Translational Equivariant Attention in Fourier Neural Operators

## Abstract

Neural operators extend conventional neural networks by expanding their functional mapping capabilities across various function spaces, thereby promoting the solving of partial differential equations (PDEs). A particularly notable method within this framework is the Fourier Neural Operator (FNO), which draws inspiration from Green's function method to directly approximate operator kernels in the frequency domain. However, after empirical observation and theoretical validation, we demonstrate that the FNO predominantly approximates operator kernels within the low-frequency domain. This limitation results in a restricted capability to solve complex PDEs, particularly those characterized by rapidly changing coefficients and highly oscillatory solution spaces. To address this challenge, inspired by the attentive equivariant convolution, we propose a novel **T**ranslational **E**quivariant **F**ourier **N**eural **O**perator (**TE-FNO**) which utilizes equivariant attention to enhance the ability of FNO to capture high-frequency features. We perform experiments on forward and reverse problems of multiscale elliptic equations, Navier-Stokes equations, and other physical scenarios. The results demonstrate that the proposed approach achieves superior performance across these benchmarks, particularly for equations characterized by rapid coefficient variations.

## 1 Introduction

Partial differential equations (PDEs) serve as a fundamental theory in scientific research, describing a wide range of physical, chemical, and biological phenomena (Sommerfeld, 1949). From turbulent fluid dynamics to atmospheric circulation and material stress analysis, many real-world phenomena are governed by PDEs. Consequently, solving PDEs is essential for tackling core challenges in the natural sciences.

Traditional numerical methods like the finite element method (FEM) and finite difference method (FDM) face significant challenges in handling noisy data, generating complex meshes, solving high-dimensional problems, and addressing inverse problems. In recent years, neural networks have emerged as powerful methods to overcome these limitations. Innovative approaches such as physics-informed neural networks (PINNs) (Karniadakis et al., 2021) and Galerkin transformers (GTs) (Cao, 2021) are specifically designed to simulate PDEs by learning mappings between inputs and outputs. Furthermore, operator-learning methods like the deep operator network (DeepONet) (Lu et al., 2019) and the Fourier Neural Operator (FNO) (Li et al., 2020; 2022) focus on learning transformations between function spaces, significantly advancing PDE-solving capabilities. Beyond solving PDEs, these neural operator-based methods have demonstrated exceptional promise in addressing complex dynamics, such as modeling climate change and predicting natural disasters (Pathak et al., 2022), highlighting their versatility and impact in real-world applications.

We focus on addressing more complex PDEs, such as multiscale PDEs, which play a pivotal role in physics, engineering, and related disciplines by enabling the analysis of intricate practical problems, including ocean circulation and high-frequency scattering (Quarteroni & Veneziani, 2003). These PDEs are characterized by rapidly varying coefficients and oscillatory solution spaces, making it essential to effectively capture information across multiple scales and frequency ranges. However, existing evidence indicates that FNO and similar methods predominantly emphasize learning low-

frequency components when solving PDEs (Liu et al., 2022; Xu et al., 2024; Liu-Schiaffini et al., 2024). This observation highlights a critical challenge: how to effectively capture high-frequency features and integrate them with low-frequency information to enhance the performance of FNO in tackling complex, multiscale PDEs.

In this paper, we propose a novel Translational Equivariant Fourier Neural Operator (TE-FNO) designed to capture and integrate low-frequency and high-frequency features across multiple scales. Drawing inspiration from attentive equivariant convolution, we theoretically demonstrate that a combined attention mechanism, comprising channel and spatial attention, can be integrated before the Fourier layer in a translationally equivariant manner. Moreover, leveraging the locally computed convolution kernel enables the efficient capture of high-frequency local details. Additionally, inspired by recent works (Xu et al., 2024; Liu-Schiaffini et al., 2024), we incorporate a Fourier kernel with a convolutional-residual layer, enhancing the model capability to effectively capture high-frequency information and address the challenges of complex multiscale PDEs. Our main contributions can be summarized as follows:

- We propose a novel TE-FNO method to address the issue where FNO-related approaches struggle to capture high-frequency features effectively. Specifically, our method integrates high-frequency and low-frequency components simultaneously with equivariant attentions and convolutional-residual Fourier layers in a hierarchical structure at various scales.

- The proposed method surpasses previous state-of-the-art approaches in existing PDE benchmarks, including Navier-Stokes equations, multiscale elliptic equations with rapidly changing coefficients, and significant solution variations.

- Furthermore, our method demonstrates effectiveness and robustness in solving inverse PDE problems, particularly when dealing with noisy input data.

## 2 RELATED WORKS

We briefly cover the background and related works in this section. More related works are listed in the Appendix A.5.

### 2.1 NEURAL PDE SOLVER

Many excellent algorithms have been proposed previously for solving PDEs using neural networks (Long et al., 2018; Hao et al., 2022). Physics-informed neural network (PINN) (Karniadakis et al., 2021) incorporates PDEs into the network by giving additional constraints with PDEs into loss function, which guide the synaptic modifications towards tuned parameters that satisfy data distribution, physical PDE laws, and other necessary boundary conditions. GT (Cao, 2021) utilizes the attention mechanism to build operator learners to solve PDEs and designs Fourier-type and Galerkin-type attention with linear complexity to reduce the computation cost. Neural operators leverage the concept that the operator denotes the mapping between infinite input and output function spaces. DeepONet (Lu et al., 2019) leverages the universal approximation theorem to derive a branch-trunk structure to form the operator in a polynomial regression way. Some other methods incorporate trained neural networks into conventional numerical solvers, to minimize numerical errors when dealing with coarse grids (Cuomo et al., 2022; Meng et al., 2020).

### 2.2 FOURIER NEURAL OPERATOR

Fourier neural operator (FNO) (Li et al., 2020; 2022) draws inspiration from the conventional Green's function method and directly optimizes the kernel within the Fourier frequency domain by utilizing the Fourier Transform. This approach has been demonstrated to be an efficient means of reducing computational cost and performing global convolution. A notable advancement is that the operator kernel is directly trained in the frequency domain, whereby the network is theoretically independent of the training data resolution. Therefore, FNO can deal with super-resolution problems and be trained on multiple PDEs.

The FNO has established a foundational framework for operator learning, inspiring several subsequent works in the field. Geo-FNO (Li et al., 2022) deforms the irregular grid into a latent space

Figure 1: Examples of various tasks. These datasets solve equations according to coefficients, previous solutions, and structures by approximating mappings between input and output in coordinate spaces. All these tasks are covered in experimental verification.

with a uniform grid to solve the limitation of Fourier Transform could only be applied to rectangular domains. F-FNO (Tran et al., 2021) learns the kernel weights in a factorized way with separable spectral layers. G-FNO (Helwig et al., 2023) utilizes the symmetry groups in the Fourier kernel to learn equivalent representations and improve accuracy even under imperfect symmetries. However, after empirical observation followed by theoretical validation, we found that FNO ignores high-frequency components by default to learn a smooth representation of the input space results in poor performance when solving PDEs with rapidly changing coefficients. (Liu-Schiaffini et al., 2024) also addressed the over-smoothing issue in FNOs, proposing localized integral and differential kernels to capture fine-grained features. In contrast to these approaches, our work focuses on designing a network architecture inspired by convolution operations and translational equivariance, specifically addressing the challenges of solving complex multiscale PDEs, rather than solely enhancing local feature extraction.

## 3 METHODS

### 3.1 PROBLEM FORMULATION

For a PDE problem, the observed samples are $(a_i, u_i)_{i=1}^N$. Assuming the coordinates in a bounded open set $\mathcal{D} \subset \mathbb{R}^d$, the input and output can be expressed as functions to these coordinates. These functions belong to the Banach spaces $\mathcal{X} = \mathcal{X}(\mathcal{D}; \mathbb{R}^{d_a})$ and $\mathcal{Y} = \mathcal{Y}(\mathcal{D}; \mathbb{R}^{d_u})$ respectively. Here, $\mathbb{R}^{d_a}$ and $\mathbb{R}^{d_u}$ denotes the range of input and output functions. $\mathcal{D}$ consists of a finite set of grid points within a rectangular area in $\mathbb{R}^2$. The function values are represented by position $x \subset \mathcal{D}$, which could be denoted as $a(x)$ and $u(x)$. The overall solving process could be viewed as using a neural network $f_\theta$ to approximate the mapping $\mathcal{X} \to \mathcal{Y}$ to predict the output $\hat{u}(x)$, where $\hat{u}(x) = f_\theta(a)(x)$.

The Fourier neural operator (Li et al., 2020) is a powerful and efficient architecture for modeling PDEs that learn operators for mapping input and output function spaces. The FNO is inspired by Green's function method by learning the kernel integral operator defined below,

$$[\mathcal{K}(\phi)a](x) := \int k_\phi(x, y)a(y)dy, \quad \forall x \in \mathcal{D}, \tag{1}$$

where Green's function kernel $k_\phi$ is learned from data and parameterized by neural networks with parameter $\phi$. FNO assumes that Green's function is periodic and only dependent on the relative distance, which means that $k_\phi(x, y) = k_\phi(x - y)$. Then the operation in Eq. 1 could be regarded as convolution and efficiently implemented as element-wise multiplication in the frequency domain by using the convolution theorem:

$$\begin{aligned}[\mathcal{K}(\phi)a](x) &:= \int k_\phi(x - y)a(y)dy \\ &= \mathcal{F}^{-1}\left(\mathcal{F}(k_\phi) \cdot \mathcal{F}(a)\right)(x) \\ &= \mathcal{F}^{-1}\left(R_\phi \cdot \mathcal{F}(a)\right)(x),\end{aligned} \tag{2}$$

where $\mathcal{F}$ and $\mathcal{F}^{-1}$ are the Fourier transform and the inverse Fourier transform, respectively. Instead of learning the kernel $k_\phi$, FNO directly learns the kernel $R_\phi$ in the Fourier domain.

## 3.2 FNO DRAWBACKS

Nevertheless, we take a one-dimensional case as an example to show that high-frequency features are not well represented in FNO and related methods, posing a challenge in dealing with multiscale PDEs. During the Fourier transformation process of the FNO, only the low-frequency components ($\omega \leq T_\omega$) are reserved for multiplication, and high-frequency components ($\omega > T_\omega$) are ignored by default. The size of the kernel is the same as the size of the reserved low-frequency components. Thus the elementwise multiplication process could be expressed as:

$$(R_\phi \cdot \mathcal{F}(a))(\omega) = \begin{cases} (R_\phi \cdot \mathcal{F}(a))(\omega), & \omega \leq T_\omega, \\ 0, & \omega > T_\omega. \end{cases}$$

Therefore, after inverse Fourier transformation,

$$\mathcal{F}^{-1}(R_\phi \cdot \mathcal{F}(a))(x) = \sum_{\omega \leq T_\omega} (R_\phi \cdot \mathcal{F}(a))(\omega)e^{i\omega x},$$

only the low-frequency components are represented. For notational convenience, we only use the one-dimensional case for illustration, which still stands for 2D and 3D cases. Additionally, previous works (Liu et al., 2022) also inform that FNO and GT have shown their tendency to prioritize learning low-frequency components before high-frequency components when applied to multiscale PDEs. Other recent works such as (Liu-Schiaffini et al., 2024) also found global operators are prone to over-smoothing and utilize local integral kernels to capture fine-grained features. The experimental result in Figure 3 (i) corroborates the high-frequency error in FNO.

## 3.3 ATTENTIVE EQUIVARIANT CONVOLUTION

In the FNO (Li et al., 2020), the kernel of Green's function is imposed as the convolution kernel, which is a natural choice from the perspective of fundamental solutions. A fundamental property of the convolution is that it commutes with translations,

$$\mathcal{L}_y [f \star \phi] (x) = [\mathcal{L}_y[f] \star \phi] (x) \tag{3}$$

where $L_y$ is the translation operator[1]. In other words, convolving a $y$-translated signal $\mathcal{L}_y[f]$ with a filter is equivalent to first convolving the original signal $f$ with the filter $\phi$ and $y$-translating the obtained response next. This property is referred to as translation equivariance.

Previous works have defined attentive group convolution (Romero et al., 2020) and proved its equivariant property. We simplify them into attentive convolution defined on $\mathbb{R}^d$,

$$[f \star^\alpha \phi] (x) = \int_{\mathbb{R}^d} \alpha(x, \tilde{x}) f(\tilde{x}) \mathcal{L}_x [\phi] (\tilde{x}) \mathrm{d}\tilde{x} \tag{4}$$

where $\alpha(x, \tilde{x})$ is the attention map between the input and output positions.

**Theorem 1.** *The attentive convolution is an equivariant operator if and only if the attention operator $\mathcal{A}$ satisfies:*

$$\forall_{\bar{x}, x, \tilde{x} \in \mathbb{R}^d} : \mathcal{A} [\mathcal{L}_{\bar{x}} f] (x, \tilde{x}) = \mathcal{A}[f] (\bar{x}^{-1} x, \bar{x}^{-1} \tilde{x}) \tag{5}$$

*If, moreover, the maps generated by $\mathcal{A}$ are invariant to one of its arguments, and, hence, exclusively attend to either the input or the output domain, then $\mathcal{A}$ satisfies Equation (5) if it is equivariant and thus, based on convolutions.*

Since convolutions and pooling operations are translation equivariant, mostly visual attention mechanisms are translation equivariant as well (Romero et al., 2020). One special case is channel attention based on fully connected layers (a non-translation equivariant map) in SE-Nets (Hu et al., 2018b). However, the input of the fully connected layers is obtained via global average pooling, which has shown that it is equivalent to a pointwise convolution (Romero et al., 2020). Therefore, attention in SE-Nets is translational equivariant as well (Cohen et al., 2018).

---

[1]It follows that $\mathcal{L}_g[f](x) = f\left(g^{-1}x\right) = f(x - y)$, where $g^{-1} = -y$ is the inverse of $g$ in the translation group $\left(\mathbb{R}^d, +\right)$ for $g = y$.

Figure 2: The overall network architecture. The input is downsampled and processed at each scale using equivariant attention and convolutional-residual Fourier layers. The final output is obtained by upsampling the outputs of various hierarchical layers.

Furthermore, previous works broadly assumed that the maps in visual attention do not depend on the filters $\phi$ and could be equivariantly factorized into spatial $\alpha^{\mathcal{X}}$ and channel $\alpha^{\mathcal{C}}$ components. Hence, the attention coefficient $\alpha$ is the sole function of the input signal and becomes only dependent on $x$.

$$[f \star^{\alpha} \phi] = [f^{\alpha} \star \phi] = [(\alpha f) \star \phi] = [(\alpha^{\mathcal{X}} \alpha^{\mathcal{C}} f) \star \phi] \tag{6}$$

In this way, the attention maps $\alpha$ can be shifted to the input feature map $f$. Resultantly, the attentive convolution is reduced to a sequence of conventional convolutions and point-wise non-linearities (Theorem 1), further decreasing attention computational cost. Furthermore, inspired by GFNO (Helwig et al., 2023), we further utilize the following theorem to establish the connection between equivariant convolution and Fourier transformation.

**Theorem 2.** *Given the orthogonal group $O(d)$ acting on functions defined on $\mathbb{R}^d$ by the map $(g, f) \mapsto L_g f$ where $(L_g f)(x) := f(g^{-1}x)$, the group action commutes with the Fourier-transform, i.e. $\mathcal{F} \circ L_q = L_q \circ \mathcal{F}$.*

This theorem describes the equivariance of the Fourier transformation, which means applying a transformation from $O(d)$ to a function in physical space equally applies the transformation to the Fourier transform of the function. Therefore, by converting Equation (6) to the frequency domain via Fourier transformation, our model could be generally built as:

$$\hat{\boldsymbol{u}}(x) := \sigma\left(\mathcal{F}^{-1}(R_\phi \cdot \mathcal{F}(\alpha^{\mathcal{X}} \alpha^{\mathcal{C}} \boldsymbol{a}))(x)\right), \ \forall x \in \mathcal{D}. \tag{7}$$

We further modify Equation (7) structure to learn the high-frequency feature better.

### 3.4 Model Architecture

We propose a method called Translational Equivariant Fourier Neural Operator (TE-FNO) which combines equivariant attention mechanisms and convolutional-residual layers to learn the function mapping at various resolutions.

**Equivariant Attention:** The attention mechanism can be conceptualized as a dynamic selection process that emphasizes significant features while suppressing irrelevant parts of the input. This mechanism has proven effective in learning dependencies among pixels in computer vision tasks (Yuan et al., 2020; Geng et al., 2021). In our context, the grid data closely resembles pixel-based image data, making attention mechanisms a natural fit for capturing dependencies within the $\mathbb{R}^d$ domain. Previous studies have extensively explored optimal configurations for combining channel and spatial attention maps in similar scenarios. Following the findings of (Woo et al., 2018), we adopt a serial approach that prioritizes channel attention before applying spatial attention, as this setup has demonstrated superior performance in prior works.

$$\tilde{\boldsymbol{a}}_c^k = \mathcal{A}^{\mathcal{C}}(\tilde{\boldsymbol{a}}^k) \odot \tilde{\boldsymbol{a}}^k, \quad \tilde{\boldsymbol{a}}_{xc}^k = \mathcal{A}^{\mathcal{X}}(\tilde{\boldsymbol{a}}_c^k) \odot \tilde{\boldsymbol{a}}_c^k, \quad \boldsymbol{v}^k = \tilde{\boldsymbol{a}}_{xc}^k + \tilde{\boldsymbol{a}}^k. \tag{8}$$

To prevent confusion, we simplify the coordinates $x$ and denote $\tilde{\boldsymbol{a}}^k$ as the input of the attention layers at the $k$-th layer, which also denotes the downsampled feature of model input $\boldsymbol{a}$. $\boldsymbol{v}^k$ represents the output of the attention layers. $\odot$ denotes the Hadamard product (also known as element-wise multiplication). The $\mathcal{A}^{\mathcal{C}}$ and $\mathcal{A}^{\mathcal{X}}$ denote the channel and spatial attention respectively and have been proved to be the equivariant operator before.

**Convolutional-Residual Fourier layers:** We propose the convolution-residual Fourier layers composed of two main components, the Fourier layer and the convolutional-residual layer. In the first component, the input feature is transformed into the frequency domain by the Fast Fourier Transform (FFT) and learning an element-wise weight kernel in the frequency domain. We follow the FNO setting, only reserving lower-frequency components and training the kernel weights on them. This setting aims to learn a smooth mapping to avoid jagged curves in solution spaces. However, this setting may ignore some details about the solution space, especially in solving multiscale PDEs. Since convolution utilizes a much smaller kernel size than the Fourier transform that allows the kernel to capture locally detailed information, we replaced the fully connected residual layers with a convolution layer. The input and output of the convolutional-residual Fourier layer at the $k$-th scale are denoted as $\boldsymbol{v}^k$ and $\tilde{\boldsymbol{v}}^k$ respectively. Thus, the Fourier layers could be modulated as:

$$\tilde{\boldsymbol{v}}^k = \sigma\left(\text{Conv}(\boldsymbol{v}^k) + \mathcal{F}^{-1}(R_\phi \cdot \mathcal{F}(\boldsymbol{v}^k))\right), \tag{9}$$

where $\sigma$ denotes the GELU activation, $R_\phi$ represents the kernel weights in the Fourier domain that should be trained. After the convolutional-residual Fourier layer, we utilize a multilayer perceptron (MLP) to integrate the feature further and obtain $k$-th layer output $\tilde{\boldsymbol{u}}^k$.

**Hierarchical architecture:** We attempt to design our model hierarchically, with various scales as inputs. As in multiscale PDEs, multiple scales and regions represent different physical laws (Karniadakis et al., 2021). The final prediction output is obtained by successively upsampling the outputs in various scales from coarse to fine. Specifically, for the $k$-scale, $\tilde{\boldsymbol{u}}^k$ is concatenated with the interpolation-upsampled $\tilde{\boldsymbol{u}}^{k+1}$ and further projected with a linear layer. More details are denoted in Appendix A.3.3.

As the weight matrix is directly parameterized in the Fourier domain, we follow the FNO (Li et al., 2020) to limit the Fourier series by terminating it at a predefined number of modes. We use different truncation values at each hierarchical layer to help the model learn diverse information at various scales. However, large truncation modes would cause computing resources to increase hugely. To balance the computation cost and performance, we set the truncation mode to decrease with the feature scale, as we reckon that large-scale features need more Fourier modes to represent. Concrete hyperparameters are presented in the Table 6.

## 3.5 EVALUATION METRICS

Previous works (Liu et al., 2022) proposed H1 loss to solve multiscale PDEs which calculates the loss in the Fourier domain. However, we only use the normalized mean squared error (N-MSE) as the loss function and evaluation metrics, which is defined as

$$\text{N-MSE} = \frac{1}{B} \sum_{i=1}^{B} \frac{\|\hat{\boldsymbol{u}}_i - \boldsymbol{u}_i\|_2}{\|\hat{\boldsymbol{u}}_i\|_2}, \tag{10}$$

where $\|\cdot\|_2$ is the 2-norm, $\boldsymbol{u}$, and $\hat{\boldsymbol{u}}$ are the ground truth and output prediction respectively.

## 4 EXPERIMENTS

**Benchmarks.** We evaluate our method on various PDE benchmarks, including multiscale elliptic equations with various resolutions, Navier-Stokes equations with different viscosity coefficients, and other physics scenarios governed by PDEs. Also, we conduct experiments on the inverse problem of multiscale elliptic equations with noise input data.

**Baselines.** We compare our method with recent and advanced methods. FNO (Li et al., 2020), U-NO (Rahman et al., 2022), and F-FNO (Tran et al., 2021) are FNO-relevant methods that use Fourier transformation to learn the operators directly in the frequency domain. WMT (Gupta et al., 2021) learns the kernel projection onto fixed multiwavelet polynomial bases. GT (Cao, 2021) modify the

Table 1: Experiment results on various elliptic equations with various resolutions. $\rightarrow$ denotes the resolution mapping between input and output. For example, $256 \rightarrow 256$ denotes the resolution of the input and the output are both $256 \times 256$. The time per epoch is measured for a batch size of 10. Performance is measured with normalized mean squared error (N-MSE with $\times 10^{-2}$). All the results of TE-FNO are averaged on 5 runs. For clarity, the best result is in **bold** and the second best is underlined. All the following tables retain this setting.

| Methods | Time per epoch (s) | Trigonometric | | Darcy Rough | | Darcy Smooth |
|---------|----|----|----|----|----|----|
| | | $256 \rightarrow 256$ | $512 \rightarrow 512$ | $128 \rightarrow 128$ | $256 \rightarrow 256$ | $64 \rightarrow 64$ |
| FNO | 7.42 | 1.936 | 1.932 | 2.160 | 2.098 | 0.83 |
| WMT | 20.03 | 1.043 | 1.087 | 1.573 | 1.621 | 0.82 |
| U-NO | 15.42 | 1.256 | 1.245 | 1.368 | 1.332 | 1.13 |
| GT | 36.32 | 1.143 | 1.264 | 2.231 | 2.423 | 1.70 |
| F-FNO | 10.42 | 1.429 | 1.424 | 1.435 | 1.513 | 0.77 |
| HANO | 29.13 | 0.893 | 0.948 | 1.172 | 1.241 | 0.79 |
| LSM | 14.26 | 0.962 | 1.093 | 1.543 | 2.368 | 0.65 |
| DCNO | 11.73 | 1.056 | 1.209 | 1.276 | **0.948** | 0.72 |
| TE-FNO | 10.81 | **0.724** | **0.699** | **1.087** | 0.963 | **0.60** |

self-attention to Galerkin-type attentions with linear complexities to solve the PDEs. HANO (Liu et al., 2022) utilizes hierarchical attention to solve multiscale PDEs. LSM (Wu et al., 2023) solves the PDEs in the latent spectrum domain by decomposing latent features into basic operators. DCNO (Xu et al., 2024) improves the structure of FNO by combining Fourier and Convolution layers.

## 4.1 MULTISCALE ELLIPTIC EQUATIONS

The elliptic equation describes the flow of fluid through a porous medium, which is formulated by a second-order linear elliptic equation,

$$\begin{cases} -\nabla \cdot (a(x)\nabla u(x)) = f(x), & x \in D, \\ u(x) = 0, & x \in \partial D, \end{cases} \tag{11}$$

with rough coefficients and Dirichlet boundary. In contrast to previous works, the coefficient functions show a significant degree of smoothness, leading to correspondingly smooth solutions. We follow the setting in DCNO (Xu et al., 2024) to change the conventional elliptic equations into multiscale cases by modifying coefficients to two-phrase rough ones (**Darcy-Rough**) or high-contrast trigonometric coefficients (**Trigonometric**). The original experiments of multiscale PDEs in (Xu et al., 2024) using coefficients with resolution $1023 \times 1023$ to approximate the solution with resolution $256 \times 256$ or $512 \times 512$, which reduce the difficulties as larger inputs might contain more specific information. To enhance the difficulty, we modify the resolution of coefficients to the same as that of the output solution. We also follow the setting in (Li et al., 2020) and perform the original elliptic equation dataset (**Darcy-Smooth**) for comparison. More details are denoted in the Appendix A.2.1.

The experimental results, presented in Table 1, demonstrate that our model consistently achieves the lowest error across various scenarios compared to other operator baselines. This is particularly evident in the case of elliptic equations with trigonometric coefficients, where the performance gap is more pronounced. Our findings suggest that cascade architecture models, such as FNO and DCNO, struggle to perform effectively in this setting. In contrast, hierarchical structures, such as U-NO and HANO, tend to deliver better results due to their inherent ability to capture multiscale dependencies. To further illustrate our model improvements in capturing high-frequency features, we provide a visualization of the predicted solutions and errors in Figure 3. Compared to FNO, our model demonstrates a significant reduction in prediction error, particularly for high-frequency components, underscoring its robustness and accuracy in challenging scenarios.

## 4.2 NAVIER-STOKES EQUATION

We consider the 2D Navier-Stokes equation, a standard benchmark proposed in FNO (Li et al., 2020), in which the vorticity forms on the two-dimensional torus $\mathbb{T}^2$. Specifically, the operator predicts the vorticity after $T_0$ by the input vorticity before $T_0$, the values of $T_0$ and $T$ vary according to the dataset. Our experiments consider viscosities with $\nu \in \{10^{-3}, 10^{-4}, 10^{-5}\}$, with smaller viscosities denoting more chaotic flow which are much harder to predict. To ensure fair comparisons,

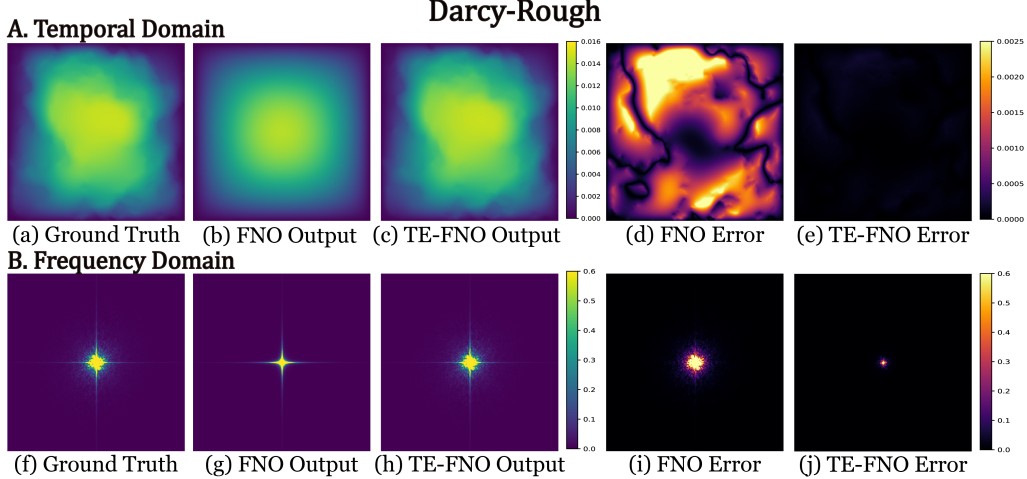

**Darcy-Rough**

**A. Temporal Domain**

(a) Ground Truth    (b) FNO Output    (c) TE-FNO Output    (d) FNO Error    (e) TE-FNO Error

**B. Frequency Domain**

(f) Ground Truth    (g) FNO Output    (h) TE-FNO Output    (i) FNO Error    (j) TE-FNO Error

Figure 3: Showcase of results on Darcy-Rough, where the high-frequency components are moved to the center The results show TE-FNO can capture more accurate high-frequency features.

Table 2: Experiments on various Navier-Stokes equations and other physical scenarios including Elasticity, and Pipe. In the Navier-Stokes dataset, the values denote the $\nu$, $T_0$, and $T$ respectively. For example, $\{10^{-3}, 10s, 50s\}$ denotes $\nu = 10^{-3}$, $T_0 = 10s$, and $T = 50s$.

| Methods | Navier-Stokes | | | Elasticity | Pipe |
|---|---|---|---|---|---|
| | $\{10^{-3}, 10s, 50s\}$ | $\{10^{-4}, 10s, 30s\}$ | $\{10^{-5}, 10s, 20s\}$ | | |
| FNO | 0.88 | 6.60 | 19.82 | 5.08 | 0.67 |
| WMT | 1.01 | 11.35 | 15.41 | 5.20 | 0.77 |
| U-NO | 0.89 | **5.72** | 17.53 | 4.69 | 1.00 |
| GT | 1.12 | 10.31 | 26.84 | 6.81 | 0.98 |
| F-FNO | 0.92 | 6.02 | 17.98 | 4.72 | 0.59 |
| HANO | 0.98 | 6.18 | 18.47 | 4.75 | 0.70 |
| LSM | 0.82 | 6.12 | 15.35 | 4.08 | **0.50** |
| TE-FNO | **0.73** | 5.87 | **15.05** | **3.91** | 0.51 |

we follow the setting in FNO, using the 'rollout' strategy to predict vorticity. The final operator could be regarded as approximated by various neural operators. More details are listed in the Appendix A.2.4.

### 4.3 OTHER PHYSICAL SCENARIOS

**Pipe**: The Pipe dataset focuses on predicting the incompressible flow through a pipe. The input is the pipe structure, while the output is the horizontal fluid velocity within the pipe. In this dataset, geometrically structured meshes with resolution $129 \times 129$ are generated. The input and output are the mesh structure and fluid velocity within the pipe.

**Elasticity**: The Elasticity dataset is designed to predict the internal stress within an incompressible material containing an arbitrary void at its center and an external tensile force is exerted on the material. Originally, the Elasticity data are presented by the point clouds, we follow (Wu et al., 2023) to modify the data into regular grids. The input consists of the material's structural characteristics, while the output represents the internal stress. More details are listed in the Appendix.

We evaluate our model on these datasets to demonstrate its effectiveness in solving general PDE problems. Table 2 summarizes experimental results on Navier-Stokes equations with various coefficients and other physical scenarios. Our model performs better in

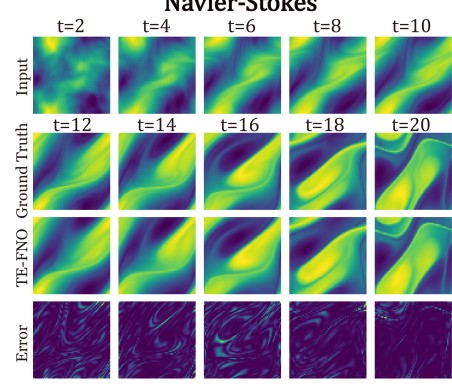

Figure 4: Showcase of results on Navier-Stokes Equation.

Table 3: Experiments on inverse coefficient identification tasks. In this experiment, the input solution space and output coefficient space are both $256 \times 256$.

| methods | Trigonometric | | | Darcy Rough | | |
|---|---|---|---|---|---|---|
| | $\epsilon = 0$ | $\epsilon = 0.01$ | $\epsilon = 0.1$ | $\epsilon = 0$ | $\epsilon = 0.01$ | $\epsilon = 0.1$ |
| FNO | 44.74 | 46.34 | 48.43 | 28.41 | 28.98 | 30.65 |
| WMT | 11.14 | 12.43 | 20.43 | 12.32 | 17.54 | 28.43 |
| U-NO | 12.97 | 18.54 | 25.87 | 15.64 | 20.54 | 25.34 |
| GT | 27.87 | 30.98 | 43.54 | 23.12 | 28.87 | 35.43 |
| F-FNO | 21.46 | 26.98 | 36.34 | 18.73 | 25.23 | 37.54 |
| HANO | 9.87 | 13.67 | 20.98 | 8.45 | 10.43 | 20.43 |
| DCNO | 8.87 | 17.64 | 34.76 | **6.32** | 11.83 | 23.54 |
| TE-FNO | **8.10** | **10.01** | **19.61** | 6.59 | **9.89** | **20.10** |

almost all settings. Learning high-frequency features may help capture detailed flow changes as flows with smaller viscosities are more chaotic. We further visualize the results of Navier-Stokes in Figure 4.

## 4.4 INVERSE PROBLEMS SOLVING

In various scientific disciplines such as geological sciences and mathematical derivation, inverse problems are of significant importance. Nonetheless, these problems frequently demonstrate reduced stability compared to their associated forward issues, even when advanced regularization techniques are employed. Following (Xu et al., 2024), we evaluate our method for inverse identification problems on multiscale elliptic PDEs. In this experiment, we aim to learn an inverse operator, which maps the solution function space to the corresponding coefficient space $\hat{u} = u + \epsilon N(u) \mapsto a$. Here, $\epsilon$ indicates the intensity of Gaussian noise introduced into the training and evaluation data. The noise term $N(u)$ accounts for the sampling distribution and data-related noise.

The experiments about inverse coefficients inference problems on the multiscale elliptic PDEs dataset are presented in Table 3. In our experiments, we modify the input and output resolutions to both $256 \times 256$ in the Trigonometric and Darcy Rough elliptic equations. Since the coefficient function space changes faster than the solution space, this task is more challenging than the forward-solving problem. The result shows that our model performs well in the inverse coefficient identification problem, which illustrates our model's ability to address the challenges posed by this ill-posed inverse problem with data. Methods such as FNO and F-FNO that learn kernel functions directly in the low-frequency domain have trouble recovering targets with high-frequency components.

## 4.5 ABLATION STUDY

To verify the effectiveness of each component in our model, we perform ablation studies in various settings, including removing components (*w/o*), replacing them with other components (*rep*), and adding some other components (*add*).

- In the *w/o* part, we consider removing the equivariant attention (*w/o* Att) and the Fourier layer (*w/o* Fourier).
- In the *rep* part, we consider replacing the convolutional-residual layer and attention mechanism with other components and keeping the number of parameters the same. For convolutional-residual layers, we replace them with simple residual layers (*rep* Conv→Res) or fully connected layers (*rep* Conv→Fc). For the attention mechanism, we replace them with dilation convolution (*rep* Att→d-Conv), multi-layer perceptrons (*rep rep* Att→MLP), self-attention (Att→SA) and adaptive token mixing (*rep* Att→ATM)
- In the *add* part, we add one hierarchical layer with corresponding attention and Fourier layer (*add* Hier).

Our ablation studies highlight the critical role of all components in our model for solving multiscale PDEs. The removal experiments show that excluding the translational-equivariant attention mechanism or the Fourier layer significantly reduces performance, demonstrating their importance. Similarly, the replacement experiments reveal that while some alternatives, like MLP, perform adequately

Table 4: Abation studies on our proposed model, including *removing components (w/o)*, *replacing them with other components (rep)*, and *adding some other components (add)*.

| Designs | | **MSE** | | |
| --- | --- | --- | --- | --- |
| | | Trigonometric | Darcy-Rough | Navier-Stokes |
| *w/o* | Attention | 1.021 | 1.110 | 18.52 |
| | FNO | 1.181 | 1.192 | 20.08 |
| *rep* | Conv→Res | 1.131 | 1.162 | 16.46 |
| | Conv→Fc | 1.176 | 1.245 | 15.72 |
| | Att→d-Conv | 0.894 | 0.993 | 16.12 |
| | Att→MLP | 0.985 | 1.034 | 15.53 |
| | Att→SA | 0.804 | 0.970 | 15.21 |
| | Att→ATM | 0.858 | 1.012 | 15.43 |
| *add* | Hier | 0.779 | **0.960** | 15.20 |
| | **TE-FNO** | **0.724** | 0.963 | **15.05** |

in simpler scenarios, they generally fail to match the effectiveness of translational-equivariant attention, especially in tasks with rapidly varying coefficients. Adding hierarchical layers may further improve the model's ability to capture multiscale features, particularly in complex scenarios, though at the cost of increased computational complexity. To balance efficiency and accuracy, we carefully limit the number of hierarchical layers in the final design. Overall, our model consistently outperforms cascade-based architectures like FNO and DCNO across benchmarks, underscoring the advantages of a translational-equivariant and hierarchical approach for capturing multiscale features and improving accuracy.

## 5 LIMITATIONS

Although TE-FNO has shown remarkable effectiveness in solving a wide range of complex PDE tasks. However, it comes with certain limitations. Firstly, TE-FNO is designed to be particularly suited to scenarios where the input and output resolutions are identical, a constraint commonly associated with U-shaped network architectures. For tasks such as input coefficient resolutions are larger than the solution resolution, TE-FNO performs suboptimal. Furthermore, TE-FNO depends on convolution-based local attention mechanisms in translational equivariant attention, while advantageous for capturing localized features, limits its ability to effectively capture global features, which can reduce its efficacy in tasks requiring a broader contextual understanding.

## 6 CONCLUSION

We propose a novel Translational Equivariant Fourier Neural Operator (TE-FNO) that combines equivariant attentions and convolutional-residual Fourier layers for solving complex PDEs. Our model utilizes equivariant attention mechanisms and convolutional residual layers to capture high-frequency features and complement the low-frequency features captured by the Fourier layer. Benefits from the above components, our model could capture both local and global features simultaneously, and at the same time, achieve superior performances in many PDE benchmarks, especially in solving forward and inverse problems of multiscale elliptic PDEs.

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

## A  Appendix

### A.1  Backgrounds and Proofs

#### A.1.1  Proofs of Equivariant of Attentive Convolution

We follow the definition of (Cohen & Welling, 2016) to define the attentive convolution and reduce the visual self-attention into

$$f_c^{out}(g) = \sum_{\tilde{c}}^{N_{\tilde{c}}} \int_G \alpha_{c,\tilde{c}}(g, \tilde{g}) \phi_{c,\tilde{c}} \left(g^{-1} \tilde{g}\right) f_{\tilde{c}}^{in}(\tilde{g}) \mathrm{d}\tilde{g}. \tag{12}$$

In this work, we only consider group act on $O(d)$, thus the definition could be further reduced into

$$f_c^{out}(x) = \sum_{\tilde{c}}^{N_{\tilde{c}}} \int_{\mathbb{R}^d} \alpha_{c,\tilde{c}}(x, \tilde{x}) \phi_{c,\tilde{c}} \left(x^{-1} \tilde{x}\right) f_{\tilde{c}}^{in}(\tilde{x}) \mathrm{d}\tilde{x}. \tag{13}$$

Without loss of generality, let $\mathfrak{A} : \mathbb{L}_2(\mathbb{R}^d) \to \mathbb{L}_2(\mathbb{R}^d)$ denote the attentive group convolution defined by Equation (13), with $N_{\tilde{c}} = N_{\tilde{c}} = 1$, and some $\phi$ which in the following we omit to simplify our derivation. Equivariance of $\mathfrak{A}$ implies that $\forall_{f \in \mathbb{L}_2(\mathbb{R}^d)}, \forall_{\bar{x}, x \in \mathbb{R}^d}$:

$$\mathfrak{A} \left[\mathcal{L}_{\bar{x}}[f]\right](x) = \mathcal{L}_{\bar{x}}[\mathfrak{d}[f]](x)$$
$$\Leftrightarrow$$
$$\mathfrak{A} \left[\mathcal{L}_{\bar{x}}[f]\right](x) = \mathfrak{A}[f] \left(\bar{x}^{-1} x\right)$$
$$\Leftrightarrow \tag{14}$$
$$\int_{\mathbb{R}^d} \mathcal{A} \left[\mathcal{L}_{\bar{x}}[f]\right](x, \tilde{x}) \mathcal{L}_{\bar{x}}[f](\tilde{x}) \mathrm{d}\tilde{x} = \int_{\mathbb{R}^d} \mathcal{A}[f] \left(\bar{x}^{-1} x, \tilde{x}\right) f(\tilde{x}) \mathrm{d}\tilde{x}$$
$$\Leftrightarrow$$
$$\int_{\mathbb{R}^d} \mathcal{A} \left[\mathcal{L}_{\bar{x}}[f]\right](x, \tilde{x}) f\left(\bar{x}^{-1} \tilde{x}\right) \mathrm{d}\tilde{x} = \int_{\mathbb{R}^d} \mathcal{A}[f] \left(\bar{x}^{-1} x, \bar{x}^{-1} \tilde{x}\right) f\left(\bar{x}^{-1} \tilde{x}\right) \mathrm{d}\tilde{x},$$

where we once again perform the variable substitution $\tilde{x} \to \bar{x}^{-1} \tilde{x}$ at the right hand side of the last step. This must hold for all $f \in \mathbb{L}_2(\mathbb{R}^d)$ and hence:

$$\forall_{\bar{x} \in \mathbb{R}^d} : \mathcal{A} \left[\mathcal{L}_{\bar{x}} f\right](x, \tilde{x}) = \mathcal{A}[f] \left(\bar{x}^{-1} x, \bar{x}^{-1} \tilde{x}\right) \tag{15}$$

#### A.1.2  Proof of Symmetry of Fourier transform to $O(d)$

Let $A \in \mathbb{R}^{d \times d}$ be an invertible matrix, $f : \mathbb{R}^d \to \mathbb{R}$ Lebesgue-integrable and $b \in \mathbb{R}^d$. Consider the function $f_{A,b} : \mathbb{R}^d \to \mathbb{R}$ given by $f_{A,b}(x) = f(Ax + b)$. Then

$$\mathcal{F}(f_{(A,b)})(\xi) = \frac{e^{-2\pi i \langle A^{-T} \xi, b \rangle}}{|\det A|} \mathcal{F}(f)(A^{-T} \xi)$$

In particular, if $A$ is an orthogonal matrix, then $|\det A| = 1$ and $A^{-T} = A$, so for all $O \in O(n)$:

$$\mathcal{F}(f_{(O,b)})(\xi) = e^{-2\pi i \langle O\xi, b \rangle} \mathcal{F}(f)(O\xi)$$

We will use the multi-dimensional change of variables formula with the substitution $z = Ax + b$, as well as the identity $\langle \xi, Az \rangle = \langle A^T \xi, z \rangle$.

$$
\begin{aligned}
\mathcal{F}(f_{(A,b)})(\xi) \\
&= \frac{1}{(2\pi)^{n/2}} \int_{\mathbb{R}^d} e^{-2\pi i \langle \xi, x \rangle} f_{(A,b)}(x)\, dx \\
&= \frac{1}{(2\pi)^{n/2} |\det A|} \int_{\mathbb{R}^d} e^{-2\pi i \langle \xi, A^{-1}(Ax+b) \rangle + 2\pi i \langle \xi, A^{-1}b \rangle} \\
&\quad f(Ax + b)\, |\det A| dx \\
&= e^{2\pi i \langle \xi, A^{-1}b \rangle} \frac{1}{(2\pi)^{n/2} |\det A|} \int_{\mathbb{R}^d} e^{-2\pi i \langle \xi, A^{-1}z \rangle} f(z)\, dz \\
&= \frac{e^{2\pi i \langle \xi, A^{-1}b \rangle}}{|\det A|} \frac{1}{(2\pi)^{n/2}} \int_{\mathbb{R}^d} e^{-2\pi i \langle A^{-T}\xi, z \rangle} f(z)\, dz \\
&= \frac{e^{2\pi i \langle A^{-T}\xi, b \rangle}}{|\det A|} \mathcal{F}(f)(A^{-T}\xi).
\end{aligned}
\tag{16}
$$

## A.2 BENCHMARK DETAILS

We introduce the underlying PDEs of each benchmark and the number of corresponding training and testing samples.

### A.2.1 MULTISCALE ELLIPTIC PDES

Multiscale elliptic equations are given by second-order linear elliptic equations,

$$
\begin{cases}
-\nabla \cdot (a(x)\nabla u(x)) = f(x) & x \in D \\
u(x) = 0 & x \in \partial D
\end{cases}
\tag{17}
$$

where the coefficient $a(x) \in [a_{\min}, a_{\max}], \forall x \in D$ and $a_{\min} > 0$. The coefficient $a(x)$, enables rapid oscillation (for example, with $a(x) = a(x/\varepsilon)$ where $\varepsilon \ll 1$), a significant contrast ratio characterized by $a_{\max}/a_{\min} \gg 1$, and even a continuum of non-separable scales.

### A.2.2 MULTISCALE TRIGONOMETRIC COEFFICIENT

We follow the setting in HANO (Liu et al., 2022), which considers eq. (17) with multiscale trigonometric coefficients. The domain $D$ is $[-1, 1]^2$, and the coefficient $a(x)$ is defined as $a(x) = \prod_{k=1}^{6} (1 + \frac{1}{2}\cos(a_k\pi(x_1 + x_2)))(1 + \frac{1}{2}\sin(a_k\pi(x_2 - 3x_1)))$, where $a_k$ is uniformly distributed between $2^{k-1}$ and $1.5 \times 2^{k-1}$, and the forcing term is fixed as $f(x) \equiv 1$. The resolution of the dataset is $1023 \times 1023$ and lower resolutions are created by downsampling with linear interpolation.

### A.2.3 TWO-PHASE COEFFICIENT

The two-phase coefficients and solutions are generated according to FNO (Li et al., 2020). The coefficients $a(x)$ are generated according to $a \sim \mu := \psi_\# \mathcal{N}\left(0, (-\Delta + cI)^{-2}\right)$ with zero Neumann boundary conditions on the Laplacian. The mapping $\psi : \mathbb{R} \to \mathbb{R}$ assigns the value $a_{\max}$ to the positive segment of the real line and $a_{\min}$ to the negative segment. The push-forward is explicitly defined on a pointwise basis. The forcing term is fixed as $f(x) \equiv 1$. The solutions $u$ are derived through the application of a second-order finite difference approach on a well-suited grid. The parameters $a_{\max}$ and $a_{\min}$ have the ability to manage the contrast of the coefficient. Additionally, the parameter $c$ regulates the roughness or oscillation of the coefficient; an increased value of $c$ leads to a coefficient featuring rougher two-phase interfaces.

### A.2.4 NAVIER-STOKES EQUATIONS

We follow the Navier-Stokes equation in FNO (Li et al., 2020). This dataset simulates incompressible and viscous flow on the unit torus, where fluid density is unchangeable. In this situation, energy

Table 5: More details about PDEs benchmarks, including the number of training and testing samples with their resolutions. NS is short for Navier-Stokes.

| Benchmarks | $N_{training}$ | $N_{testing}$ | Resolution |
|---|---|---|---|
| Trigonometric | 1000 | 100 | $512 \times 512, 256 \times 256$ |
| Darcy-Rough | 1000 | 100 | $256 \times 256, 128 \times 128$ |
| Darcy-Smooth | 1000 | 200 | $64 \times 64$ |
| NS($\nu = 10^{-3}$) | 1000 | 200 | $64 \times 64$ |
| NS($\nu = 10^{-4}$) | 10000 | 2000 | $64 \times 64$ |
| NS($\nu = 10^{-5}$) | 1000 | 200 | $64 \times 64$ |
| Elasticity | 1000 | 200 | $41 \times 41$ |
| Pipe | 1000 | 200 | $129 \times 129$ |

conservation is independent of mass and momentum conservation.

$$\nabla \cdot u = 0$$
$$\frac{\partial w}{\partial t} + u \cdot \nabla w = \nu \nabla^2 w + f \tag{18}$$
$$w|_{t=0} = w_0,$$

where $u$ and $w$ are abbreviated versions of $u(x, t)$ and $w(x, t)$, respectively. $u \in \mathbb{R}^2$ is a velocity vector in 2D field, $w = \nabla \times u$ is the vorticity, $w_0 \in \mathbb{R}$ is the initial vorticity at $t = 0$. In this dataset, viscosity is set as $\nu \in \{10^{-3}, 10^{-4}, 10^{-5}, 10^{-6}\}$ and the resolution of the 2D field is $64 \times 64$. The number of training and prediction frames is varied in different settings.

### A.2.5 ELASTICITY

The governing equation of Elasticity materials is:

$$\rho^s \frac{\partial^2 \boldsymbol{u}}{\partial t^2} + \nabla \cdot \boldsymbol{\sigma} = 0, \tag{19}$$

where $\rho^s \in \mathbb{R}$ denotes the solid density, $\nabla$ and $\boldsymbol{\sigma}$ denote the nabla operator and the stress tensor respectively. Function $\boldsymbol{u}$ represents the displacement vector of material over time $t$. These benchmarks estimate the inner stress of incompressible materials with an arbitrary void in the center of the material. In addition, external tension is applied to the material. This benchmark's input and output are the material's structure and inner stress.

### A.2.6 PIPE

This dataset focuses on the incompressible flow through a pipe. The governing equations are similar to Navier-Stokes equations:

$$\nabla \cdot \boldsymbol{U} = 0$$
$$\frac{\partial \boldsymbol{U}}{\partial t} + \boldsymbol{U} \cdot \nabla \boldsymbol{U} = \boldsymbol{f} - \frac{1}{\rho} \nabla p + \nu \nabla^2 \boldsymbol{U}. \tag{20}$$

In this dataset, geometrically structured meshes with resolution $129 \times 129$ are generated. The input and output are the mesh structure and fluid velocity within the pipe.

We provide details of our benchmarks including the number of training and testing samples and their input solutions in Table 5.

## A.3 MODEL DETAILS

### A.3.1 IMPLEMENTATION DETAILS.

Our model is implemented in PyTorch and conducted on a single NVIDIA A100 40GB GPU. Here are the implementation details of our model.

Table 6: Model configurations.

| MODEL DESIGNS | HYPERPARAMETERS | VALUES |
|---|---|---|
| FOURIER LAYERS | LOW-FREQUENCY MODES $\{d_{low}^1, \cdots, d_{low}^K\}$ | $\{24, 12, 6, 3\}$ |
| HIERARCHICAL LAYERS | CHANNELS OF EACH SCALE $\{d_c^1, \cdots, d_c^K\}$ | $\{32, 64, 128, 128\}$ |
| | NUMBER OF SCALES $K$ | 4 |
| | DOWNSMAPLE RATIO $r$ | 2 |
| TRAINING SETTING | LEARNING RATE | 0.0005 |
| | BATCHSIZE | 10 |

### A.3.2 MODEL CONFIGURATIONS.

Here we present the detailed model configurations of our model in Table 6. In the beginning, the input data will be padded with zeros properly to resolve the division problem in model configurations.

### A.3.3 DOWNSAMPLE AND UPSAMPLE ARCHITECTURE

In this section, we illustrate the downsampling and upsampling operations in our hierarchical architectures. Our method is similar to that of LSM (Wu et al., 2023).

**Downsampling.** Given deep features $\{\tilde{a}^k(x)\}_{x \in \mathcal{D}^k}$ at the $k$-th scale, The downsampling operation is to aggregate deep features in a local region through maximum pooling and convolution operations, which can be formalized as:

$$\{\tilde{a}^{k+1}(x)\}_{x \in \mathcal{D}^{k+1}} = \text{Conv}\left(\text{MaxPool}\left(\{\tilde{a}^k(x)\}_{x \in \mathcal{D}^k}\right)\right),$$
$$k \text{ from } 1 \text{ to } (K-1). \tag{21}$$

**Upsampling.** Given the features $\tilde{u}^{k+1}(x)_{x \in \mathcal{D}^{k+1}}$ and $\tilde{u}^k(x)_{x \in \mathcal{D}^k}$ corresponding to the $(k+1)$-th and $k$-th scales, respectively, the upsampling procedure involves fusing the interpolated features from the $(k+1)$-th scale and the features from the $k$-th scale using local convolution. This process can be expressed as follows:

$$\{\hat{u}^k(x)\}_{x \in \mathcal{D}^k} = \text{Conv}\left(\text{Concat}\left(\left[\text{Interp}\left(\{\tilde{u}^{k+1}(x)\}_{x \in \mathcal{D}^{k+1}}\right), \{\tilde{u}^k(x)\}_{x \in \mathcal{D}^k}\right]\right)\right),$$
$$k \text{ from } (K-1) \text{ to } 1, \tag{22}$$

where we adopt the bilinear Interpolation $\text{Inter}(\cdot)$ for 2D data.

### A.4 MORE VISUALIZATION RESULTS

We visualize more results compared to FNO on the Trigonometric dataset in Figure 5.

### A.5 MORE RELATED WORK

### A.5.1 OPERATOR LEARNING

Suppose $\mathcal{A}$ and $\mathcal{U}$ denote the infinite input and output function spaces. The objective of the operator is to learn a mapping from $\mathcal{A}$ to $\mathcal{U}$ using a finite collection of input and output pairs in the supervised learning way. For any vector function $a \in \mathcal{A}$, $a : \mathcal{D}_{\mathcal{A}} \to \mathbb{R}^{d_{\mathcal{A}}}$ with $\mathcal{D}_{\mathcal{A}} \subset \mathbb{R}^d$ and for any vector function $u \in \mathcal{U}$, $u : \mathcal{D}_{\mathcal{U}} \to \mathbb{R}^{d_{\mathcal{U}}}$, with $\mathcal{D}_{\mathcal{U}} \subset \mathbb{R}^d$. Given =the training data $\{(a_i, u_i)\}_{i=1}^N$, our objective is to train an operator $G_\theta : \mathcal{A} \to \mathcal{U}$ which is parameterized by $\theta$, to learn the mapping between input and output function spaces by extracting relationships from $a$ and $u$.

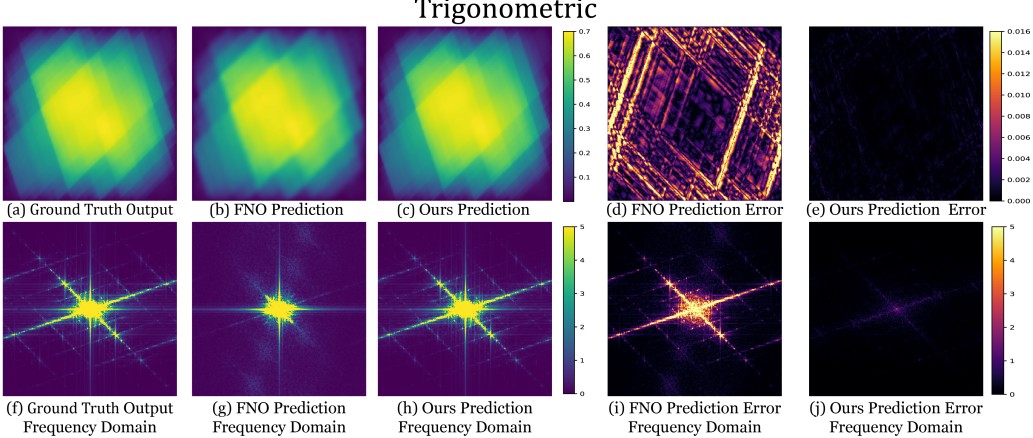

Figure 5: Showcase of Trigonometric Elliptic PDEs, where the high-frequency components are moved to the center.

### A.5.2 MULTISCALE PDES

Multiscale PDEs have many applications, including forecasting atmospheric convection and ocean circulation, modeling the subsurface of flow pressures (Furman, 2008; Huyakorn, 2012), the deformation of elastic materials (Rivlin, 1948; Merodio & Ogden, 2003), and the electric potential of conductive materials (Sundnes et al., 2005). Multiscale elliptic PDEs are classic examples of multiscale PDEs. Solving elliptic PDEs with smooth coefficients is a conventional problem that can be effectively addressed using FNO. However, when the coefficients become non-smooth and exhibit rapidly changing features, the values in the solution spaces can exhibit oscillations and high contrast ratios (Xu et al., 2024). Another example is turbulent flow, which is modeled by the Navier-Stokes equation. This equation describes fluid dynamics and exhibits turbulent behavior at high Reynolds numbers. In turbulent flow, unsteady vortices interact, leading to complex dynamics. To solve these multiscale PDEs effectively, models must account for both global and local dynamics.

### A.5.3 NUMERICAL SOLVERS FOR MULTISCALE PDES

In addressing multiscale PDEs, a variety of numerical approaches are available. Numerical homogenization methods (Engquist & Souganidis, 2008) aim to create a finite-dimensional approximation space for solution exploration. Fast solvers like multilevel and multigrid methods (Hackbusch, 2013; Xu & Zikatanov, 2017) can be considered as an extension of numerical homogenization. Recently, operator-adapted wavelet methods, such as Gamblets (Owhadi, 2017), have been developed to solve linear PDEs with rough coefficients, representing a progression beyond numerical homogenization. Nevertheless, handling multiscale PDEs poses inherent challenges for numerical methods, given that the computational cost tends to scale inversely proportional to the finest scale $\epsilon$ of the problem. In recent years, there has been increasing exploration of neural network methods for solving multiscale PDEs (Liu et al., 2022).

### A.5.4 MORE FNO RELATED WORK

Networks inspired by FNO have been verified in various domains, including computer vision and time series forecasting (Ovadia et al., 2023a;b). AFNO (Guibas et al., 2021) leverages kernel in the Fourier domain as a token mixer within the transformer, aiming at reducing computational complexity and enhancing performance in segmentation tasks. FEDformer (Zhou et al., 2022) harnesses sparse basic elements in the Fourier frequency domain to create a frequency-enhanced transformer. Meanwhile, GFNet (Rao et al., 2021) employs the element-wise multiplication of learnable global filters with features in the frequency domain to improve the performance in classification and transfer learning tasks.

### A.5.5 Visual Attention Methods

The attention mechanism can be regarded as a process of adaptive selection based on input features. It has yielded advantages in numerous visual tasks, including image classification (Woo et al., 2018), object detection (Dai et al., 2017; Hu et al., 2018a), and semantic segmentation (Yuan et al., 2020; Geng et al., 2021). In computer vision, the attention mechanism is usually be divided into three main categories: channel attention, spatial attention, and temporal attention. For instance, SENet (Hu et al., 2018b) utilizes global average pooling on the channel dimension to modulate the corresponding channel attention. Complementary channel attention akin to that of CBAM (Woo et al., 2018) and BAM (Park et al., 2018) utilize similar strategies for spatial attention and combine spatial and channel attention in series and parallel respectively. Recent research in visual attention aims to integrate the strengths of various attention mechanisms to create more holistic attention (Hu et al., 2018b; Guo et al., 2022).

