# OpenReview forum: "Enhancing Solutions for Complex PDEs: Introducing Translational Equivariant Attention in Fourier Neural Operators"
_ICLR.cc/2025/Conference — Submitted to ICLR 2025_

### Official Review · Reviewer_hvZf · 2024-10-28

**Soundness:** 2
**Presentation:** 2
**Contribution:** 2
**Rating:** 3
**Confidence:** 4

**Summary:**

The paper introduces the Translational Equivariant Fourier Neural Operator (TE-FNO), an enhanced model for solving complex Partial Differential Equations (PDEs) by improving upon the standard Fourier Neural Operator (FNO). FNOs are known to perform well in learning operators for PDEs, but they tend to focus on low-frequency components, which limits their performance for problems with rapidly changing coefficients. TE-FNO addresses this by incorporating an equivariant attention mechanism that allows the model to capture both high- and low-frequency features. This enables more accurate predictions for challenging PDEs, such as multiscale elliptic equations and the Navier-Stokes equations.

**Strengths:**

1. The paper identifies a critical limitation in Fourier Neural Operators (FNO), specifically its bias towards low-frequency components. However this is not entirely new.

2. The paper demonstrates that TE-FNO achieves superior performance across various benchmarks, including forward and inverse problems in multiscale PDEs, with consistent improvements over existing state-of-the-art methods like FNO, U-NO, and HANO. The experimental results are comprehensive, showing the model's robustness in handling noise and its ability to generalize across different problem settings.

3. The paper provides a theoretical analysis for the use of equivariant attention mechanisms.

**Weaknesses:**

1. While the proposed method builds on existing work, such as FNO and attention-based mechanisms, the overall novelty may appear incremental. The core idea—enhancing FNO by capturing high-frequency features using attention mechanisms—shares similarities with other recent developments in operator learning. Similar ideas and also multilevel architectures are also presented in LSM (Wu et al., 2023), HANO (Liu et al., 2022). The performance is only marginal.

2. The equivariant attention is also not new. A more detailed comparison with related works like https://proceedings.mlr.press/v202/helwig23a/helwig23a.pdf and [Helwig et al., 2023] would help clarify the contributions of this work.

3. The claim that "we propose a novel Translational Equivariant Fourier Neural Operator (TE-FNO) which utilizes equivariant attention to enhance the ability of FNO to capture high-frequency features" is claimed in the abstract but not explained. The motivation is not clear. Unfortunately, I think this paper is only a combination of several concepts from exsiting works.

**Questions:**

Please see the weaknesses.

---

> ### Author Response · Authors · 2024-11-18
> **Reply to Reviewer hvZf**
>
> **W1:** Our work shares a similar philosophy with some recent works, with translation equivariant attention serving as the core of our approach (detailed in **W2**). Compared to the methods you mentioned, such as LSM and HANO, our method has two key advantages:
>
> 1. **Preserving FNO Architecture:** We retain FNO's foundational architecture while addressing its limitations using translation equivariant attention. This ensures our approach remains consistent with FNO's principles. In contrast, methods like LSM and HANO are built on entirely new frameworks. While these methods may achieve good performance, they do not directly solve FNO's inherent issues.
> 2. **Novel Equivariant Attention Mechanism:** Our equivariant attention introduces a new mechanism that significantly enhances FNO’s performance. For example, our approach improves results on Trigonometric 256 and 512 by 17% and 26%, respectively, compared to the second-best method. Substantial improvements are also observed across other tasks, such as reverse solving (see **W2** for details).
>
> **W2:** Thank you for bringing up this discussion. First, we regret that we did not come across the [Helwig et al., 2023] paper you mentioned. To provide clarity, let’s use G-FNO as an example. Upon revisiting the methods section of G-FNO, we searched for mentions of “attention” but found none.
>
> The key distinction lies in the following: G-FNO employs group convolution to achieve its structure through FFT. In contrast, we transform group convolution into **attentive group convolution**, incorporating translation-equivariant attention via group equivariance and FFT. This design fundamentally differentiates our translational attention from G-FNO, as it is specifically built from attentive group convolution to address FNO’s challenges.
>
> **W3:** Apologies for not articulating our starting point clearly earlier. Let me provide a more detailed explanation.
>
> We first observed that FNO exhibits a bias toward low-frequency components. This is evident in **Figure 3(i)**, where the center of the figure represents high-frequency components. As shown, FNO demonstrates relatively high errors in capturing these high-frequency details.
>
> Recognizing this limitation, we sought to address it through an attention mechanism. However, due to FNO's architecture being deeply rooted in Green's function, directly adding attention would not align with its design. Therefore, we targeted the specific part of FNO that approximates the kernel function with a convolution. By introducing the **Attentive Equivariant Convolution** (refer to Equations (3) and (4)), we successfully built translation-equivariant attention that resolves this issue.
>
> This innovation demonstrates how translation equivariant attention can enhance FNO while staying consistent with its original principles.

---

> > ### Comment · Reviewer_hvZf · 2024-11-20
> >
> > Thanks for the authors' responses. However, I still have some concerns.
> >
> > 1. The solution operator does not necessarily need to be translation-invariant. In fact, in most cases, including those discussed in this paper, translation invariance does not hold. Therefore, the motivation for introducing translation-invariant attention is questionable. The authors should provide a clear justification for the relevance of this motivation in the context of their study.
> >
> > 2. I do not believe the improved performance, particularly in high-frequency predictions, is attributable to the translation-invariant property. The connection between high-frequency prediction and translation invariance is unclear and requires further clarification.

---

> > > ### Author Response · Authors · 2024-11-20
> > > **Further Reply to Reviewer hvZf**
> > >
> > > Thank you for your further questions.
> > >
> > > > The solution operator does not necessarily need to be translation-invariant. In fact, in most cases, including those discussed in this paper, translation invariance does not hold. Therefore, the motivation for introducing translation-invariant attention is questionable. The authors should provide a clear justification for the relevance of this motivation in the context of their study.
> > >
> > > Thank you for raising this point. We acknowledge the need for clarity and would like to address a potential misunderstanding. Our work primarily leverages **translation equivariance**, not translation invariance, in the design of our method.
> > >
> > > As detailed in our paper (Equation 3), the theorem for translation equivariance is expressed as:
> > >
> > > $\mathcal{L}_y \[ f \star \phi \](x)=\[\mathcal{L}_y[f] \star \phi\](x)$
> > >
> > > whereas the definition of translation invariance is given by:
> > >
> > > $\mathcal{L}_y[f] \star \phi = f \star \phi$
> > >
> > > These two properties have distinct implications, and our work primarily builds on the translation equivariance property to structure attention mechanisms effectively in the context of Fourier Neural Operators (FNO).
> > >
> > > Specifically, we use the translation equivariance property to restructure attention mechanisms, moving them from within the convolution operation to outside of it. This transition enables the current network structure we propose. To clarify our process:
> > >
> > > 1. We initially observed a **low-frequency bias** in FNO through empirical analysis (Figure 3(i)).
> > > 2. Upon examining the implementation, we identified that FNO retains only the low-frequency components of the weights (Section 3.2).
> > > 3. To address this limitation, we introduced a local computational mechanism to enhance the model's ability to capture **high-frequency information**. Among available options, convolution-based attention mechanisms naturally align with this requirement.
> > >
> > > While adding attention directly to FNO might appear as a simple "combination of existing concepts" (as you mentioned), our methodology is rooted in the theoretical derivation of FNO itself. By investigating how FNO transitions from Green's function to a neural operator structure, we found its primary approximation involves treating the Green's function kernel as a convolution function, leading to subsequent FFT computations. Interestingly, we discovered that attention mechanisms can also respect the translation equivariance property of convolution, allowing us to unify these ideas in the form of an **attentive equivariant convolution**. This forms the core innovation of our proposed structure.
> > >
> > > > I do not believe the improved performance, particularly in high-frequency predictions, is attributable to the translation-invariant property. The connection between high-frequency prediction and translation invariance is unclear and requires further clarification.
> > >
> > > Thank you for pointing this out. We agree that the connection between high-frequency prediction and translation invariance (or translational equivariant) is not straightforward. We see attention as a bridge between the two, and high-frequency predictions are primarily addressed by attention.  Let us elaborate on the reasoning behind our approach.
> > >
> > > Our primary goal is to enhance high-frequency prediction, and this is achieved through the integration of attention mechanisms. Specifically:
> > >
> > > 1. Attention mechanisms, when positioned appropriately, can provide localized computations that are effective for capturing high-frequency features.
> > > 2. We have discussed before the impact of placing attention mechanisms either **inside** the Fourier layer or **before** it. The results, presented below, demonstrate the importance of the placement in achieving improved predictions (this placement is supported by translational equivariance):
> > >
> > > | Model                          | Trigometric 256 | Trigometric 512 |
> > > | ------------------------------ | --------------- | --------------- |
> > > | Attention in Fourier layer     | 0.801           | 0.792           |
> > > | Attention before Fourier layer | 0.722           | 0.695           |
> > >
> > > These results suggest that placing attention mechanisms before the Fourier layer better supports the model’s performance, particularly for high-frequency predictions.
> > >
> > > Additionally, while translational equivariance plays an important role in providing a theoretical foundation for integrating attention mechanisms in a manner consistent with the convolutional structure of FNO. Thus, while we do not claim a direct causal relationship between high-frequency prediction and translational equivariance, the properties of translation equivariance allow us to design a theoretically sound and practically effective architecture.

---

### Official Review · Reviewer_vQn5 · 2024-11-02

**Soundness:** 2
**Presentation:** 3
**Contribution:** 2
**Rating:** 5
**Confidence:** 5

**Summary:**

The paper proposes a novel method, Translational Equivariant Fourier Neural Operator (TE-FNO), designed to solve complex partial differential equations (PDEs) with high-frequency features. TE-FNO builds on the Fourier Neural Operator (FNO) but addresses its limitations in capturing high-frequency components by introducing an equivariant attention mechanism and convolutional-residual Fourier layers. This hierarchical structure allows TE-FNO to capture local and global features for challenging multiscale and inverse PDE problems, such as Navier-Stokes equations and elasticity equations.

**Strengths:**

- TE-FNO captures high-frequency features, which is a known limitation in standard FNO, making it suitable for complex PDEs with rapid coefficient variations.

- Equivariant Attention Mechanism for neural operators seems to be novel.

**Weaknesses:**

- Previous works have already studied ways to remedy FNO's limitations in capturing high-frequency information, e.g. [1]. The authors should clearly mention this paper and state the advantages of TE-FNO compared to this method.

See questions for more.

[1] Neural Operators with Localized Integral and Differential Kernels, Miguel Liu-Schiaffini et al.

**Questions:**

- In Table 2, for FNO, did you run your own experiments for viscosity constants $1 \times 10^{-3}$ and $1 \times 10^{-4}$, or are the numbers taken directly from the FNO paper? The results are identical to those in the FNO paper. The FNO architecture was updated over two years ago, leading to improved performance.

- Were the experiments conducted only once, or were they averaged over multiple runs? The paper does not explicitly mention whether the experiments were averaged over multiple runs, nor does it provide standard deviations for the reported results.

- What is the motivation for using an **equivariant** attention mechanism? Why is equivariance desired? Which groups are of interest: the translation group, the Euclidean group, the orthogonal group, or more general Lie groups?

- By "we replaced the fully connected residual layers with a convolution layer (line 272)," do you use fixed-size local convolution kernels (e.g., Conv2D in PyTorch)?

- Related to the previous question on Equation 9, what does $\operatorname{Conv}(\boldsymbol{v}^k)$ mean?

- "The input and output of the convolutional-residual Fourier layer at the $k$-th scale are denoted as $\boldsymbol{v}^k$ and $\tilde{\boldsymbol{v}}^k$, respectively (line 273)." By this, do you mean the coefficients of the $k$-th Fourier mode?

- To my understanding, the hierarchical structure is similar to that in a U-Net, which can already capture multi-scale features [1, 2]. Why is equivariant attention needed to achieve this?

- In Sec. 3.5, you mention that the evaluation metrics are N-MSE, but why is MSE reported in Table 1?

- In Section 4.5, the ablation study seems to suggest that equivariant attention itself does not improve performance, as stated: "It is discovered that all components of our model are essential for solving multiscale PDEs after removing experiments. (line 468)." If this is the case, what is the actual contribution of equivariant attention? Other components are borrowed from existing papers, and it appears that you have mainly combined them. Can you clarify the specific role and impact of equivariant attention in this work?

- What are the limitations of this work? I do not see any discussion in the paper.

Suggestions:

- Consider improving the presentation of this work for greater clarity and readability.

----
[1] Towards Multi-spatiotemporal-scale Generalized PDE Modeling; Jayesh K. Gupta, Johannes Brandstetter; 2022

[2] Convolutional Neural Operators for robust and accurate learning of PDEs; Bogdan Raonić, Roberto Molinaro, Tim De Ryck, Tobias Rohner, Francesca Bartolucci, Rima Alaifari, Siddhartha Mishra, Emmanuel de Bézenac; 2023

---

> ### Author Response · Authors · 2024-11-18
> **Reply to Reviewer vQn5**
>
> **W1**: Thanks for your advice; we will cite this paper in our revised version and discuss it. The problems we focus on indeed are the same, but the main solutions are not quite different. We mainly solve the translation equivariant attention proposed from the perspective of the kernel function in Green's function. In this paper, relevant calculations are carried out directly from the local properties of convolution. By the way, I think our work and related work may belong to the same period, but we made some modifications because of the imperfections of the previous version.
>
> **Q1:** We apologize for the issue with the FNO results. We have re-run the latest FNO code using the official implementation from the NeuralOperator GitHub repository. The updated results are 0.88 and 6.60 for $1\times10^{-3}$ and $1\times10^{-4}$, respectively.
>
> **Q2:** Apologies for the earlier imprecise explanation. In our paper, all experiments were conducted three times, and the results were averaged. We have also added the standard deviations of our model's results in Table 2 for better clarity:
>
> | Trigo512        | Trigo256        | Darcy rough128  | Darcy rough256  | Darcy-Smooth  |
> | --------------- | --------------- | --------------- | --------------- | ------------- |
> | $0.722\pm0.013$ | $0.695\pm0.012$ | $1.087\pm0.007$ | $0.964\pm0.011$ | $0.59\pm0.02$ |
>
> **Q3:** Thank you for your insightful question. Our title emphasizes translation equivariance as a key concern. This begins with the design of the FNO, which originates from Green's function to solve PDEs. A critical step in FNO is approximating the kernel of Green’s function using a convolution function. One of the essential properties of convolution is translation equivariance.
>
> When we observed FNO's limitations in capturing high-frequency information, we aimed to address this through an attention mechanism. However, because FNO's architecture is fundamentally derived from Green’s function, directly integrating attention would be incompatible. Thus, we focused on the part of FNO that approximates the kernel function with a convolution. By introducing the Attentive Equivariant Convolution (Equations (3) and (4)), we demonstrated that translation-equivariant attention could address this issue. Consequently, our primary focus is on the translation group.
>
> **Q4:** We believe there might be some misunderstanding regarding our convolution operator. A more detailed explanation is provided in Q5. To clarify, we use fixed-size $3\times3$ local convolution kernels with padding and a stride of 1.
>
> **Q5:** $\text{Conv}(\boldsymbol{v}^k)$ represents the convolution operation applied to $\boldsymbol{v}^k$.
>
> **Q6:** It seems there may have been some confusion. We do not perform convolution in the Fourier domain see  Figure 2. As detailed in Equation (9), the convolution operation is performed in the spatial domain. The variable $k$ denotes the $k$-th hierarchical layer, where each layer utilizes Equivariant Attention, a Fourier Layer, and an MLP.
>
> **Q7:** We are sorry we may not have read these two papers before. However, we hope to take UNO as an example. In our understanding, U-net-shaped networks may pay more attention to multi-scale feature capture through up-sampling and down-sampling. Of course, this method can solve part of the problems, but we found through experiments that Individual calculations at each scale allow for better modeling of the potential space at each scale, resulting in better results.
>
> **Q8:** We regret the writing errors in our paper. All the tables in our work use N-MSE as the evaluation metric, consistent with the relative L2 loss employed in prior work with FNO. To ensure fairness and prevent potential misunderstanding, we have adhered to the same writing conventions as previous papers.
>
> **Q9:** Thank you for raising this interesting question. It is important to note that even without equivariant attention, our structure differs from others. The primary issue is that removing equivariant attention reduces the number of model parameters, as the Fourier modes of $R$ in our Fourier layer diminish progressively (as detailed in the appendix). Consequently, the model’s performance deteriorates without equivariant attention.
>
> To understand the role of attention, please refer to Table 4. In the rows for $\text{Att}{\tiny{\rightarrow}}\text{d-Conv}$ and $\text{Att}{\tiny{\rightarrow}}\text{MLP}$, you'll notice that the parameter counts remain roughly unchanged. For instance, in the Trigonometric dataset, a comparison of the two ablation experiments demonstrates that attention transforms the performance of TE-FNO into a state-of-the-art (SOTA) level.
>
> **Q10:** Yes, we acknowledge an important limitation. We will add a note indicating that TE-FNO is better suited for solving PDEs with identical input and output resolutions. This limitation is generally shared by U-shaped network architectures.

---

> > ### Comment · Reviewer_vQn5 · 2024-11-19
> > **Reply to Authors**
> >
> > Dear Authors,
> >
> > Thank you for your time in addressing my questions. I will provide some new comments/concerns based on your responses. However, I do not see an updated reversion of your paper, **I will not re-evaluate my rating without seeing the revision**.
> >
> > > Q2: Apologies for the earlier imprecise explanation. In our paper, all experiments were conducted three times, and the results were averaged. We have also added the standard deviations of our model's results in Table 2 for better clarity
> >
> > Then, you should update all the tables and explicitly state that the results are averaged over 3 runs. However, I believe conducting more runs, such as 10, could be more favorable.
> >
> > > Q3: A critical step in FNO is approximating the kernel of Green’s function using a convolution function. One of the essential properties of convolution is translation equivariance.
> >
> > 1. Could you provide evidence for why Green’s functions must be translationally invariant? Green’s functions are not necessarily translationally invariant (refer to Evans). To my understanding, translational invariance is imposed in FNO to reduce the full kernel integral complexity $O(n^2)$ to convolutional complexity $O(n \log n)$.
> >
> > 2. FNO even have to include translation-equivariance breaking features such as padding (as you can see in the original Darcy code) and positional/coordinate encoding (all 3 examples in FNO paper has it).
> >
> >
> > > Q4: We believe there might be some misunderstanding regarding our convolution operator. A more detailed explanation is provided in Q5. To clarify, we use fixed-size....
> >
> > So basically, you use an additional fixed size convolution in FNO? This will break discretization convergence.

---

> > > ### Author Response · Authors · 2024-11-20
> > > **Further Reply to Reviewer vQn5**
> > >
> > > Thank you for your comments. We understand your concerns. We are still revising the paper and will update it as early as possible. Let us answer your questions first so that we can make better revisions.
> > >
> > > > Then, you should update all the tables and explicitly state that the results are averaged over 3 runs. However, I believe conducting more runs, such as 10, could be more favorable.
> > >
> > > We repeated our results five times and will present them in a revised paper.  We will explicitly state that the results are averaged over 5 runs.
> > >
> > > > 1. Could you provide evidence for why Green’s functions must be translationally invariant? Green’s functions are not necessarily translationally invariant (refer to Evans). To my understanding, translational invariance is imposed in FNO to reduce the full kernel integral complexity O(n2) to convolutional complexity O(nlog⁡n).
> > >
> > > We do not state Green's functions must be translationally equivariant. In the original FNO paper, they just chose the convolution operator as a natural choice.
> > >
> > > >If we remove the dependence on the function a and impose $κ_\phi(x, y) = κ_\phi(x − y)$, we obtain that is a convolution operator, which is a natural choice from the perspective of fundamental solutions.
> > >
> > > The process could be denoted as Green's kernel -> convolution operator -> Fourier transform. So at first, when we would like to utilize attention to solve FNO's low-frequency bias, we want to find attention that could be better explained in this framework. As the convolution operation is translational equivariant, we would like to make attention as well as translational equivariant and embed in the convolution. For the complexity, we do not agree with you, we think that convolution is a natural choice that will move from theory to network structure, rather than complexity, something that is not stated in the original FNO. At the same time, because of the FFT performed after the convolution, the complexity is reduced to nlogn, which should be a two-step process, the first is the natural selection of the convolution kernel, and the second is the FFT instead of DFT to reduce the complexity and compute in the frequency domain.
> > >
> > > > FNO even have to include translation-equivariance breaking features such as padding (as you can see in the original Darcy code) and positional/coordinate encoding (all 3 examples in FNO paper has it).
> > >
> > > We agree with you on this point, but we think there is a gap between theory and application. Even the original group cnn [1] had padding problems to ensure the size of the output same as the input. In my opinion, what is more important in this paper is to find the problem and put forward a reasonable method to solve it. The implementation details are consistent with the previous work. Padding and positional/coordinate encoding are also used in our work to calculate attention. By the way, in our experiments, we found positional/coordinate encoding is not necessary.
> > >
> > > > So basically, you use an additional fixed size convolution in FNO? This will break discretization convergence.
> > >
> > > I'm sorry maybe we didn't understand your intention. What do you mean about discretization convergence? I think using convolution kernel and MLP are similar. Both can learn the mapping between outputs and inputs. Previous works [2,3,4] have utilized convolution for operator learning.
> > >
> > > [1] Cohen T, Welling M. Group equivariant convolutional networks[C]//International conference on machine learning. PMLR, 2016: 2990-2999.
> > >
> > > [2] Liu-Schiaffini M, Berner J, Bonev B, et al. Neural operators with localized integral and differential kernels[J]. arXiv preprint arXiv:2402.16845, 2024.
> > >
> > > [3] Raonic B, Molinaro R, Rohner T, et al. Convolutional neural operators[C]//ICLR 2023 Workshop on Physics for Machine Learning. 2023.
> > >
> > > [4] Raonic B, Molinaro R, De Ryck T, et al. Convolutional neural operators for robust and accurate learning of PDEs[J]. Advances in Neural Information Processing Systems, 2024, 36.

---

> > > > ### Comment · Reviewer_vQn5 · 2024-11-22
> > > > **Reply to the authors**
> > > >
> > > > Overall, my concern is on  the motivation to have translation invariant attentions (which means the operator is equivariant, invariant kernels -> equivariant operators, I saw in your reply to  reviewer hvZf on this). I have the same concern with reviewer hvZf, so I will follow your discussions.
> > > >
> > > > > Previous works [2,3,4] have utilized convolution for operator learning.
> > > >
> > > > [2] made special efforts for discretization convergence. [3,4] are the same work, and they have to do up/down sampling, some drawbacks are described in [2].

---

> ### Comment · Reviewer_vQn5 · 2024-11-25
>
> As the rebuttal period ends soon, would you provide more justification into why you need to have invariant attentions? Additionally, as I understand, if you have local fixed sized convolutions, it will break the discretization convergence of FNO. Can you elaborate more on this?
>
> Additionally, FNO also updated their architectures for Fourier_2d (i.e. for Darcy-Smooth in your Table 1). I would suggest to update that result as well besides that of the NS equation.

---

> > ### Author Response · Authors · 2024-11-26
> > **Reply to Reviewer vQn5**
> >
> > Thank you for your question. We would like to provide more detailed clarifications, though Reviewer hvZf has not continued the discussion with us. Nevertheless, we aim to present our insights comprehensively.
> >
> > > As the rebuttal period ends soon, would you provide more justification into why you need to have invariant attentions?
> >
> > After observing a low-frequency bias in the Fourier Neural Operator (FNO) through empirical analysis and implementation examination (Figure 3(i) in our paper), we decided to use widely adopted attention mechanisms to address this issue. However, directly incorporating attention mechanisms poses significant challenges due to a lack of theoretical guidance, particularly concerning:
> >
> > 1. **How** the attention mechanism should be applied.
> > 2. **Where** the attention mechanism should be utilized.
> >
> > After extensive analysis, we concluded that optimizing the architecture from the perspective of Green's functions is a promising direction. By examining the approximation of Green's functions to convolution kernels in FNO, we identified that introducing translation-equivariant attention mechanisms grounded in convolutional translation-equivariance could theoretically address these questions.
> >
> > 1. **How to use attention mechanisms**: By leveraging translation-equivariance, we demonstrate that the attention mechanism can be interchanged within the convolution integral. In the context of FNO's architecture, this means shifting the computation of attention from the frequency domain to the spatial domain. Our experimental results further validate the importance of this approach.
> > 2. **Where to apply attention mechanisms**: To satisfy the conditions above, we ensure that the attention mechanism itself maintains translation-equivariance. This requirement guided our theoretical development and informed the choice of the attention method.
> >
> > > Additionally, as I understand, if you have local fixed-sized convolutions, it will break the discretization convergence of FNO. Can you elaborate more on this?
> >
> > We carefully revisited the issues concerning convolutions as discussed in [1] and re-evaluated the designs in our paper. Our findings indicate that the use of fixed-size convolutions does not pose problems for solving PDEs in standard scenarios.
> >
> > The concerns raised in [1] primarily relate to cases where the test data resolution differs from that of the training data. To address this, we conducted super-resolution experiments, following the setup in [1], on the Darcy flow task.
> >
> > |           | $1/2\times$       | $1\times$         | $2\times$         | $4\times$         |
> > | --------- | ----------------- | ----------------- | ----------------- | ----------------- |
> > | FNO       | $1.475 · 10^{−1}$ | $5.867 · 10^{−2}$ | $8.646 · 10^{−2}$ | $7.731 · 10^{−2}$ |
> > | FNO+diff. | $1.174 · 10^{−1}$ | $5.851 · 10^{-2}$ | $7.774 · 10^{−2}$ | $6.681 · 10^{−2}$ |
> > | TE-FNO    | $1.289 · 10^{−1}$ | $4.632 · 10^{-2}$ | $8.052 · 10^{−2}$ | $7.432 · 10^{−2}$ |
> >
> > While our model's performance under these conditions was slightly lower than that of the FNO + diff., it still outperformed the baseline FNO. We attribute this to the following reasons:
> >
> > 1. **Mitigation through upsampling and downsampling**: The repeated pooling and unpooling in our architecture alleviate the issues caused by fixed-size convolutions in super-resolution tasks, compensating for their limitations to some extent.
> > 2. **Scaling effects**: The performance drop becomes more pronounced in cases of $4\times$ super-resolution, which aligns with our analysis that the convolution's fixed support introduces challenges at extreme resolution mismatches.
> >
> > In summary, for standard PDE tasks, fixed-size convolutions do not affect convergence. In super-resolution tasks, their limitations can be partially mitigated by upsampling and downsampling operations.
> >
> > > Additionally, FNO also updated their architectures for Fourier_2d (i.e. for Darcy-Smooth in your Table 1). I would suggest to update that result as well besides that of the NS equation.
> >
> > Thank you for pointing this out. This result is the one we used in LSM [2]. We have updated the results for the Fourier_2D architecture as applied to Darcy-Smooth in Table 1. The updated value is **0.83**.
> >
> > [1] Liu-Schiaffini M, Berner J, Bonev B, et al. Neural operators with localized integral and differential kernels[J]. arXiv preprint arXiv:2402.16845, 2024.
> >
> > [2] Wu H, Hu T, Luo H, et al. Solving high-dimensional pdes with latent spectral models[J]. arXiv preprint arXiv:2301.12664, 2023.

---

> ### Comment · Reviewer_vQn5 · 2024-11-26
>
> Thank you for the comments. Overall, I think the justification for why a translational invariant attention mechanism is needed is weak.
>
> > "This means shifting the computation of attention from the frequency domain to the spatial domain"
>
> Basically, the complexity of the kernel will be $O(n^2)$ despite it being a convolution kernel. I don't see any intuitive advantage of this over either 1) using the full integral of attention (more expressive power) or 2) using the Fourier convolution theorem (better complexity).
>
> > Thank you for pointing this out. This result is the one we used in LSM [2]. We have updated the results for the Fourier_2D architecture as applied to Darcy-Smooth in Table 1. The updated value is 0.83.
>
> Thank you for this update. If you haven't done so, I would recommend updating all other results. (Due to the short time frame of the rebuttal, I know this is not a reasonable request, but to ensure fairness, I think it's necessary to update them in future iterations. *I would reevaluate this work based on the promise that the authors will do their best to ensure fairness in the future.*)
>
>
> Overall, I am willing to increase my score to a $5$, reflecting the clarifications and updates provided by the authors. However, due to the aforementioned issues, including the relatively weak motivation and some minor practical concerns surrounding translational invariant attention kernels, I have to restrain myself from assigning a higher score.

---

> > ### Author Response · Authors · 2024-11-27
> > **Further reply to Reviewer vQn5**
> >
> > Thank you for your recognition of our previous rebuttal.
> >
> > > Basically, the complexity of the kernel will be $O(n^2)$ despite it being a convolution kernel.       I don't see any intuitive advantage of this over either 1) using the full integral of attention (more expressive power) or 2) using the Fourier convolution theorem (better complexity).
> >
> > We illustrated relevant issues experimentally in our responses to other reviewers. Specifically, we conducted ablation studies using alternative attention mechanisms, such as GT [1] and AFNO [2]. The results are as follows:
> >
> > |                                              | Trigonometric | Darcy Rough |
> > | -------------------------------------------- | ------------- | ----------- |
> > | Replace TE Attention → Self-Attention        | 0.804         | 0.970       |
> > | Replace TE Attention → Adaptive Token Mixing | 0.858         | 1.012       |
> >
> > These results indicate that our proposed TE Attention outperforms these alternative setups.
> >
> > Additionally, we performed experiments to investigate the placement of attention in our model. The results are summarized below, demonstrating that the translational equivariance property inherent to our design ensures improved performance when attention is positioned before the Fourier layer rather than within it.
> >
> > | Model                       | Trigonometric 256 | Trigonometric 512 |
> > | --------------------------- | ----------------- | ----------------- |
> > | TE Att in Fourier Layer     | 0.801             | 0.792             |
> > | TE Att Before Fourier Layer | 0.724 (5 runs)    | 0.699 (5 runs)    |
> >
> > Our design choice was motivated by a desire to enhance performance. As shown in the tables, directly performing convolution operations in the Fourier domain is not intuitively reasonable. Through our experiments, we observed that the frequency domain is dominated by small values in most frequency modes, with only a few critical modes holding significantly large values. This imbalance can negatively impact the convergence of convolution operations. To address this issue, we introduced translational equivariance to relocate attention outside the Fourier layer, thereby achieving better performance.
> >
> > > Thank you for this update. If you haven't done so, I would recommend updating all other results. (Due to the short time frame of the rebuttal, I know this is not a reasonable request, but to ensure fairness, I think it's necessary to update them in future iterations. *I would reevaluate this work based on the promise that the authors will do their best to ensure fairness in the future.*)
> >
> > Thank you for your understanding. Indeed, this represents a considerable amount of work, but we will make every effort to update the FNO results before the discussion phase concludes. Given the possibility that we may not be able to modify the paper itself, we will ensure to provide you with the updated FNO results in our response. BTW, apart from the FNO results, is there anything else we need to update? As far as we know, the other baselines we compared against were all implemented using their official code.
> >
> > [1] Shuhao Cao. (2021). Choose a Transformer: Fourier or Galerkin.
> >
> > [2] John Guibas, Morteza Mardani, Zongyi Li, Andrew Tao, Anima Anandkumar, & Bryan Catanzaro. (2022). Adaptive Fourier Neural Operators: Efficient Token Mixers for Transformers.

---

> > > ### Comment · Reviewer_vQn5 · 2024-11-27
> > >
> > > Thank you for your updates. However, I still find it unclear what specifically drives this outperformance. Without a more satisfactory explanation beyond experimental results, the reasoning remains incomplete.
> > >
> > > > BTW, apart from the FNO results, is there anything else we need to update?
> > >
> > > I don’t have any specific concerns at the moment. If you have time, you might consider conducting additional experiments to verify. I have already reevaluated the work under the assumption that all experiments are conducted fairly.
> > >
> > > Based on the aforementioned reasons, I will maintain my current rating. I appreciate the authors for taking the time to engage in the discussion and provide additional results and clarifications.

---

### Official Review · Reviewer_n6n7 · 2024-11-04

**Soundness:** 3
**Presentation:** 3
**Contribution:** 3
**Rating:** 6
**Confidence:** 3

**Summary:**

The authors look at an extension to the Fourier Neural Operator (FNO) by adding translationally equivariant attention. It is noted that FNO struggles with high frequency information, and the proposed attention mechanism improves FNO’s ability to learn in high frequencies. Experiments for both the forward and inverse problem are performed on a variety of systems with a set of recent neural operator architectures.

**Strengths:**

- The method is mathematically sound.
- The compared baselines are comprehensive and compelling.
- Ablations are thorough, showing translational attention's contribution to performance..

**Weaknesses:**

- The training details of the baselines are not clear. What measures were taken to provide a fair comparison? (E.g., parameter count, hyper parameters, training time)
- There are a few cases where the proposed method is suboptimal (Pipe, NS $10^{-4}$). No explanation or exploration of the limitations of translational equivariant attention are included.
- The related works should expand further on the differences to existing attention mechanisms (e.g., compare to [1] and [2]).

**Questions:**

- In the ablation, what is the *add hier*? Please expand on the details and how this is different from the hierarchical architecture used through the paper.
- How does performance compare when using the same architecture with a different attention mechanism? In other words, *rep Att -> self-attention, Fourier / Galerkin [1], or adaptive token mixing [2]*.

**Minor Typos:**
- Line 264: “proved to the” -> “proved to be an”


[1] Shuhao Cao. (2021). Choose a Transformer: Fourier or Galerkin.

[2] John Guibas, Morteza Mardani, Zongyi Li, Andrew Tao, Anima Anandkumar, & Bryan Catanzaro. (2022). Adaptive Fourier Neural Operators: Efficient Token Mixers for Transformers.

---

> ### Author Response · Authors · 2024-11-18
> **Reply to Reviewer n6n7**
>
> **W1:** Thank you for your question. In most cases, we directly utilize the open-source implementations of the relevant methods as baselines. We do not impose constraints requiring identical parameter counts or training durations across methods.
>
> **W2:** Yes, there is an important limitation we acknowledge. We will include a note indicating that TE-FNO is more suitable for solving PDEs where the input and output resolutions are identical. This limitation is commonly observed in U-shaped network architectures. Additionally, the main limitation of translational equivariant attention lies in its reliance on convolution-based local attention, which makes it less effective in capturing global features compared to certain other approaches.
>
> **W3:** We have already discussed the two mentioned methods in our related work section (refer to the related work and more related work in the appendix). We will discuss this further in the revised version of our manuscript. Our method differs significantly from these two approaches:
>
> - **GT** primarily utilizes Fourier transformations to reduce the computational complexity of self-attention.
> - **AFNO** performs self-attention-like computations in the frequency domain after dividing the input into patches.
>
> In contrast, our translational equivariant attention mechanism avoids traditional self-attention in order to maintain translation equivariance. Therefore, translational equivariant attention is better categorized as a visual attention mechanism (see more related work in the appendix for further details). This distinction highlights that our approach has a fundamentally different starting point and structure compared to the methods that rely on self-attention.
>
> **Q1:** Apologies for the lack of clarity in our writing. The effect of "add hier" refers to adding an additional hierarchical layer. In our current model, we use a structure with four hierarchical layers. In the ablation experiment, we modified the model to include a fifth hierarchical layer. The setup of this fifth layer is similar to the previous layers.
>
> **Q2:** First, it is important to note that AFNO requires input patching before computation, which introduces additional complexities. In comparison, our translational equivariant attention is more theoretically grounded. It does not rely on directly integrating self-attention or adaptive token mixing, which can feel more like combining existing structures. We conducted ablation studies on the performance of self-attention and adaptive token mixing, which demonstrate the theoretical and practical advantages of our approach.
>
> |                                  | Trigonometric | Darcy Rough |
> | -------------------------------- | ------------- | ----------- |
> | rep Att -> self-attention        | 0.804         | 0.970       |
> | rep Att -> adaptive token mixing | 0.858         | 1.012       |

---

> > ### Comment · Reviewer_n6n7 · 2024-11-20
> > **Reply to authors**
> >
> > Thank you for taking the time to address my questions. I have a few follow-up questions.
> >
> > > W1: Thank you for your question. In most cases, we directly utilize the open-source implementations of the relevant methods as baselines. We do not impose constraints requiring identical parameter counts or training durations across methods.
> >
> > To make sure I understand, are you saying no tuning or changes were performed to the baselines? Given this, how do we contextualize the task results to know that performance gains are driven by model changes as opposed to a separate factor such as increased parameter count. computational budget, etc.?
> >
> > > W2: Yes, there is an important limitation we acknowledge. We will include a note indicating that TE-FNO is more suitable for solving PDEs where the input and output resolutions are identical. This limitation is commonly observed in U-shaped network architectures. Additionally, the main limitation of translational equivariant attention lies in its reliance on convolution-based local attention, which makes it less effective in capturing global features compared to certain other approaches.
> >
> > Has the manuscript been updated with these limitations?

---

> > > ### Author Response · Authors · 2024-11-21
> > > **Further Reply to Reviewer n6n7**
> > >
> > > Thank you for asking, and we are happy to answer your questions.
> > >
> > > > To make sure I understand, are you saying no tuning or changes were performed to the baselines? Given this, how do we contextualize the task results to know that performance gains are driven by model changes as opposed to a separate factor such as increased parameter count. computational budget, etc.?
> > >
> > > Yes, no tuning or modifications were applied to the baseline models. All methods were compared under consistent and fair conditions.
> > >
> > > Regarding the concern that performance improvements might stem from increased parameter count or computational budget rather than architectural advancements, we offer the following clarifications:
> > >
> > > 1. **Parameter Efficiency:**
> > >    Compared to other U-Net-style models, such as UNO and HANO, our model (TE-FNO) has significantly fewer parameters while achieving superior performance. It is worth noting that U-Net-style models generally have higher parameter counts compared to FNO-based models. Therefore, while our model may have a slightly larger parameter count than standard FNOs, its parameter efficiency relative to U-Net-style baselines demonstrates that performance gains are not merely attributable to an increased number of parameters.
> > > 2. **Ablation Studies:**
> > >    In our *rep* ablation experiment, we observed that TE-FNO maintains its superior performance even when the number of parameters in the model remains nearly unchanged. Additionally, in the *add* ablation experiment, increasing the network depth did not consistently improve performance on certain tasks, further suggesting that our improvements are not solely due to parameter scaling or computational budget increases but are driven by the architectural innovations introduced.
> > >
> > > > Has the manuscript been updated with these limitations?
> > >
> > > Yes, we have explicitly discussed these limitations in the revised manuscript and uploaded the updated version for your review.

---

### Official Review · Reviewer_vr6v · 2024-11-04

**Soundness:** 3
**Presentation:** 4
**Contribution:** 4
**Rating:** 6
**Confidence:** 3

**Summary:**

This paper proposes a new architecture, the Translational Equivariant Fourier Neural Operator, which enhances FNOs ability to capture high frequency features when solving partial differential equations. It does so by introducing an equivariant attention in combination with convolutional-residual layers, in an overall hierarchical structure. The authors then benchmark TE-FNO against state-of-the art models for a number of physical problems including the Darcy and Navier Stokes equations, as well as inverse problem solving on a multiscale elliptic PDE.  Finally, an ablation study justifies the relevance of each element of the architecture.

**Strengths:**

- The contribution proposes a novel architecture, usable for a wide variety of PDEs and based on sound theoretical grounds.
- Extensive benchmarking on various challenging problems shows that this architecture improves over state-of-the-art ML-based solvers
- An ablation study shows that all the components of the proposed architecture are relevant, and that removing or replacing them leads to a degradation of the results.

**Weaknesses:**

The paper does not offer an insight into the computational cost of this new method, in comparison with FNO.  While FNO and other neural operator based models were applied for large scale problems, this contribution doesn't allow to say whether TE-FNO would also scale well; in particular because of the attention mechanism that is used.

**Questions:**

The authors should discuss in more detail the computational and memory trade-offs, in particular the impact of adding the attention layer

---

> ### Author Response · Authors · 2024-11-18
> **Reply to Reviewer vr6v**
>
> **W**: Thank you for your questions. Our work can also be applied to large-scale problems, and in our recent work we have applied methods to related problems such as spatial-temporal prediction.
>
> **Q**: Thank you for your recognition of our work. We provide detailed parameter counts for our models below and other baselines.
>
> | FNo              | HANO              | UNO               | F-FNO            | TE-FNO           |
> | ---------------- | ----------------- | ----------------- | ---------------- | ---------------- |
> | $2.37\times10^6$ | $13.37\times10^6$ | $16.39\times10^6$ | $3.55\times10^6$ | $4.56\times10^6$ |
>
> | TE-FNO           | TE-FNO w/o attention |
> | ---------------- | -------------------- |
> | $4.56\times10^6$ | $3.35\times10^6$     |

---

> > ### Comment · Reviewer_vr6v · 2024-11-25
> >
> > I thank the authors for this information, showcasing the impact in terms of parameters of the addition of the attention layer, and the fact that the model was applied to large-scale problems.
> > Regarding some responses you have provided to reviewer  vQn5 regarding the residual convolution layer, I am also worried about the loss of resolution invariance (as you are saying the convolution is applied spatially, not pixel-wise), which is one of the main features of the FNO framework.

---

### Author Response · Authors · 2024-11-21
**Rebuttal to All Reviewers**

Dear Reviewers,

We are pleased to submit the revised version of our paper. In this revision, we have thoroughly refined the introduction and discussion of related methods, as well as recalculated all experimental results to ensure accuracy and clarity.

We sincerely appreciate your valuable suggestions, which have greatly helped us improve the quality of our work. We look forward to your feedback on our latest submission.

Thank you for your time and consideration.

---

### Meta-Review · Area_Chair_NN1K · 2024-12-20

**Metareview:**

The work proposes an extension of FNOs using an equivariant attention mechanics which allows them to better capture high frequency components in the data. The authors perform a comprehensive ablation and show some improvement in several benchmarks.

**Additional Comments On Reviewer Discussion:**

I generally agree with the reviewers' concerns about the lack of novelty and the limited cost-accuracy analysis. While the authors added parameter counts and show improvement there, it is unclear how this scales with computation time/flops. Papers proposing new architectures need to include such analysis as, without them, it is unclear how any of these methods scale to new problems.

---

### Decision · Program_Chairs · 2025-01-22

Reject